# The perceived impact of homelessness on health during pregnancy and the postpartum period: A qualitative study carried out in the metropolitan area of Nantes, France

**Giulio Borghi**[1]*, **Pascal Caillet**[2], **Sylvaine Devriendt**[1], **Maxime Lebeaupin**[2], **Maud Poirier**[3], **Juan-Diego Poveda**[4]

1 Regional Delegation of Pays de la Loire, Médecins du Monde, Nantes, France, 2 Public Health Department, CHU of Nantes, Nantes Université, Nantes, France, 3 Department of Gynecology and Obstetrics, CHU of Nantes, Nantes Université, Nantes, France, 4 Research and Knowledge Management Unit, Médecins du Monde, Saint Denis, France

* giulio.borghi@chu-nantes.fr

## Abstract

The number of homeless people has been constantly increasing in Europe over recent years, as well as the proportion of women among the homeless population. Pregnancy can increase the risk of becoming homeless and, on the other hand, homelessness has been widely connected to adverse perinatal outcomes. The objective of this study was to describe the overall perceived impact of homelessness on health during pregnancy and the postpartum period, using a qualitative research approach to prioritize women's perspective. One-time semi structured interviews were conducted with 10 pregnant women and 10 women in the postpartum period experiencing homelessness in the metropolitan area of Nantes, as well as with six people from their social surroundings. A thematic analysis was performed to identify major themes and sub-themes. Homelessness was perceived as having an overall negative impact on all aspects of health (physical health, mental health, and social well-being) during pregnancy and the postpartum period. Stress and anxiety, food insecurity, social isolation, physical suffering, deterioration of chronic diseases, and pregnancy complications, were the main perceived consequences of homelessness on health. On the other hand, social support, and the "welcomeness" of healthcare professionals during pregnancy and the postpartum period were identified as capable of palliating those consequences. Finally, basic needs, such as having access to suitable housing, being independent, and being in good health, were identified by participants in the study as their main priorities. The results of this study, as well as those found by previous research, allowed us to identify possible axes in tackling homelessness and its complex consequences on health during pregnancy and the postpartum period. Housing and income assistance interventions, promoting social support and employment, outreach services enhancing collaborative networks among healthcare service providers, and integrating coordinated multidisciplinary approaches in primary care have shown to provide promising solutions to this issue.

**Data Availability Statement:** The minimal data needed to support our study results have been integrated in the "Results" section of the paper. To assure and protect participant's privacy, the full datasets (interview transcriptions) are not available. In fact, due to the qualitative methodology adopted in the study, interviews may contain sensitive information that could lead to participants identification. Moreover, the authors have assured data privacy and confidentiality to participants before enrolling on the study, assuring that interview transcripts would only be overseen for research purposes by the authors.

**Funding:** The authors received no specific funding for this work.

**Competing interests:** The authors have declared that no competing interests exist.

## Introduction

Estimating the number of homeless people is a challenging task, mainly because of the heterogeneous definition of homelessness and the limitations in the capability and methodology of data collection on this phenomenon [1]. However, an overall rise in the number of homeless people has been described in multiple EU Member States [1]. In the specific case of France, the last estimations made by the French Court of Auditors, had shown an increase in homelessness of nearly 110% between 2012 (143,000 people) and 2019 (300,000 people) [2].

Housing conditions have a major impact on health, making the nature of housing one of the key social determinants of health [3]. Previous studies showed that homeless people are more frequently affected by health problems than populations with similar characteristics but properly housed [4–7]. These problems concerned all aspects of health (physical health, mental health, and social-well-being). The relationship between housing conditions and the health status of people is complex, notably because homelessness can cause health problems and, conversely, a poor health status can be the starting point of homelessness [6, 8].

Despite the majority of homeless people still being males, the proportion of women is increasing among the homeless population, reaching nearly 40% [1, 9]. Regarding women, pregnancy can increase the risk of homelessness and homelessness has been related to complications during pregnancy and the postpartum period [10]. Compared to the general population, newborns of homeless women have higher probabilities of being small for gestational age, have a lower birth-weight, are more often born preterm, are hospitalized for longer, require more intensive care, and receive less breastfeeding [11].

Although several studies have already explored the impact of homelessness on health during pregnancy and the postpartum period, a comprehensive description of this phenomenon, prioritizing women's perspective, would be a helpful tool in identifying possible solutions. As a matter of fact, most of the documented evidence available focuses only on specific dimensions of homelessness (e.g., housing instability) or on specific health outcomes (e.g., perinatal outcomes), and does not provide a general picture of this phenomenon from the point of view of the directly concerned population. The objective of our study was to describe how the health of homeless women who are pregnant or in the postpartum period is impacted, according to them, by their physical, social, and healthcare surroundings.

## Methods

A qualitative approach was chosen, to provide a comprehensive description of the phenomenon studied and to prioritize the perspective of the study sample. The consolidated criteria for reporting qualitative research (COREQ) were used to detail research methods and results [12]. These criteria are summarized in a checklist of items that should be included in reporting qualitative research, providing the page number on which each specific item is addressed (see checklist in S1 Table). This study was reviewed and approved by the local ethics committee "Groupe Nantais d'Éthique dans le Domaine de la Santé" (GNEDS).

### Participants and recruitment

The inclusion criteria adopted were the following: i) being a homeless pregnant woman or woman in the postpartum period (6 weeks following childbirth) at the time of recruitment or being a member of the social surroundings (family members and friends) of a homeless pregnant woman or woman in the postpartum period, ii) being over 18 years-old or being a minor with an attested consent to participate in the study from a parent or a legal guardian, and iii)

not being affected by a psychiatric disorder incompatible with the correct understanding of the research questions.

The choice to include people from the social surroundings of the homeless pregnant women or women in the postpartum period was made to promote a global vision of the perceived impact of homelessness on the health of the study sample, and to reinforce the robustness of the results by cross-referencing the data collected during interviews. The data obtained from this specific subset of the population was used to complement and to enrich the information already provided by homeless pregnant women or women in the postpartum period, being careful, nevertheless, to respect women's priorities and perspective.

We used the European Typology of Homelessness and housing exclusion (ETHOS) 'Light' developed by the European Federation of National Organizations Working with the Homeless (FEANTSA) to define the dimensions of homelessness included as a selection criteria in this research [13]. This definition aims to cover all living situations that can be considered as forms of homelessness across Europe: rooflessness, houselessness, living in insecure housing, and living in inadequate housing. Only pregnant women or women in the postpartum period experiencing one of these living situations were considered to meet the inclusion criterion of being homeless, and could, therefore, be recruited.

Recruitment was organized through purposive sampling and conducted directly in the field (non-conventional dwellings, healthcare institutions, homeless hotels, overnight shelters, etc.). The sites of recruitment were determined with the help of the Nantes delegation of Médecins du Monde, health professionals of the Nantes University Hospital (CHU of Nantes), and other associations working with homeless people. During field visits, people meeting the inclusion criteria were informed about the research's objectives and goals. Informed oral consent was provided by all participants in the study, witnessed by the interviewer and by the professional interpreter (when present), and documented on a dedicated form, as suggested by the GNEDS.

## Data collection

Data was collected through one-time individual semi-structured interviews. Field notes regarding the setting and course of interviews, as well as interviewer perceptions and thoughts

**Table 1. Interview guide used for homeless pregnant women or homeless women in the postpartum period.**

| Pre-identified category | Questions |
|---|---|
| 1. Impact of homelessness on health. | 1.1 *What does "being in good health" mean to you*? |
| | 1.2 *How do you think that your living conditions impact or influence your health in a positive or negative way*? |
| | 1.3 *What helped you to take care of yourself and your child during and after pregnancy*? |
| | 1.4 *What are you missing to properly take care of yourself and of your child during and after pregnancy*? |
| 2. Pregnancy follow-up during homelessness. | 2.1 *Could you explain how your pregnancy follow-up is going/ went*? |
| | 2.2 *What are the factors that influenced your pregnancy follow-up in a positive or negative way*? |
| 3. Role of the social surroundings during pregnancy and the postpartum period. | 3.1 *Could you tell me about the role and involvement of people from your social surroundings during pregnancy and after delivery*? |
| | 3.2 *How do your social surroundings impact in a positive or negative way your health*? |
| 4. Priorities of homeless women during pregnancy and the postpartum period. | 4.1 *If today you could change something, what would be your priority and why*? |

were collected during the whole interview, while socio-demographic characteristics of the study population were collected using a questionnaire at the end of each interview. Professional interpreters were engaged to allow the recruitment and participation of people who were not completely fluent in French. Of the 26 interviews, nine were conducted in French, eight in Susu, seven in Romanian, one in Fula, one in English, and one in Portuguese.

The interview guide (Table 1) contained open-ended questions organized under the following pre-identified categories of interest: i) impact of homelessness on health, ii) pregnancy follow-up during homelessness, iii) role of the social surroundings during pregnancy and the postpartum period, and iv) priorities of homeless women during pregnancy and the postpartum period. The guide was developed with the aim of promoting the participants' point of view and to explore the dimensions they prioritized. If necessary, follow-up questions were prepared to further explore the themes that emerged during interviews. A second interview guide (Table 2) was created to reformulate and to adapt questions to the members of the social surroundings, keeping the same organization of the interview around the previously listed pre-identified categories.

All the interviews were conducted by GB, a male medical doctor specialized in public health, working for Médecins du Monde, who was previously introduced to and trained in qualitative research. No previous relationship between the interviewer and the participants existed. This study was the first research experience on the topic of homelessness for the interviewer. Regular feedbacks were organized with a qualitative research specialist (JDP) to provide methodological support when needed. Interviews were conducted directly on the field or in the offices of Médecins du Monde, being careful to assure participants' privacy and comfort. All interviews were audio-recorded, made anonymous, and then transcribed by GB. Interviews lasted on average 36 minutes (from 18 minutes, for the shortest, to 54 minutes for the longest).

## Data analysis

Interviews were conducted, transcribed, and coded by GB without following a specific scheme. He performed a thematic content analysis of interviews using Microsoft Excel. Continuous thematization was used to construct the coding tree, starting from the key categories pre-identified

**Table 2. Interview guide used for members of the social surroundings of homeless pregnant women or homeless women in the postpartum period.**

| Pre-identified category | Questions |
|---|---|
| 1. Impact of homelessness on health. | 1.1 *What does "being in good health" mean to you?* |
|  | 1.2 *How do you think that the living conditions of your friend/sister/wife/. . . impact or influence her health in a positive or negative way?* |
|  | 1.3 *What helps her to take care of herself and of her child during and after pregnancy?* |
|  | 1.4 *What do you think she misses to properly take care of herself and of her child during and after pregnancy?* |
| 2. Pregnancy follow-up during homelessness. | 2.1 *Could you explain how her pregnancy follow-up is going/went?* |
|  | 2.2 *What are the factors that influenced her pregnancy follow-up in a positive or negative way?* |
| 3. Role of the social surroundings during pregnancy and the postpartum period. | 3.1 *Could you tell me about the role and involvement of people from her social surroundings during pregnancy and after delivery?* |
|  | 3.2 *How do her social surroundings impact in a positive or negative way her health?* |
| 4. Priorities of homeless women during pregnancy and the postpartum period. | 4.1 *If today something could change for her, what would be the priority and why?* |

**Table 3. Socio-demographic characteristics of the study sample.**

| Qualitative variables | Pregnant women | Women in the postpartum period | Members of the social surroundings |
|---|---|---|---|
| | N (%) | N (%) | N (%) |
| **Feels surrounded by family or friends** | | | |
| *Yes* | 5 (50%) | 8 (80%) | - |
| *No* | 5 (50%) | 2 (20%) | - |
| **Marital status** | | | |
| *Single* | 4 (40%) | 5 (50%) | 0 (0%) |
| *Married or in a couple* | 6 (60%) | 5 (50%) | 6 (100%) |
| **French language** | | | |
| *Does not speak French* | 2 (20%) | 1 (10%) | 0 (0%) |
| *Understands French, but has difficulty speaking it* | 3 (30%) | 2 (20%) | 1 (17%) |
| *Speaks French, but cannot read or write* | 4 (40%) | 2 (20%) | 2 (33%) |
| *Speaks French, can read and write* | 1 (10%) | 5 (50%) | 3 (50%) |
| **Transport** | | | |
| *No public or private transport* | 1 (10%) | 0 (0%) | 0 (0%) |
| *Access to transport with difficulties* | 3 (30%) | 1 (10%) | 2 (33%) |
| *Access to transport without difficulties* | 6 (60%) | 9 (90%) | 4 (67%) |
| **Health coverage** | | | |
| *No medical coverage* | 7 (70%) | 3 (30%) | 3 (50%) |
| *Medical coverage without complementary insurance* | 1 (70%) | 1 (10%) | 1 (17%) |
| *Medical coverage with complementary insurance* | 2 (20%) | 6 (60%) | 2 (33%) |
| **Pregnancy follow-up** | | | |
| *Yes* | 8 (80%) | 10 (100%) | - |
| *No* | 2 (20%) | 0 (0%) | - |
| **Respondent status** | | | |
| *Couple* | - | - | 4 (67%) |
| *Cousin* | - | - | 1 (17%) |
| *Sister-in-law* | - | - | 1 (17%) |
| **Quantitative variables** | Median (Interquartile range) | Median (Interquartile range) | Median (Interquartile range) |
| **Age** (years) | 21.0 (3.3) | 26.0 (5.0) | 31.5 (15.3) |
| **Month of pregnancy** (months) | 7.0 (3.0) | - | - |
| **Postpartum day** (days) | - | 28.0 (12.8) | |
| **Month of first pregnancy follow-up visit** (months) | 3.5 (1.3) | 3.0 (2.5) | - |

in the interview guides (Tables 1 and 2) and enriching it with themes that emerged from the interviews. Meetings with the other two authors (SD and JDP), who had sound experience with the study sample and qualitative research methods, were regularly organized to discuss findings and to rearrange the coding tree. The coding tree remained unchanged after the 20th interview, and data collection stopped when theoretical saturation was achieved at the 26th interview. The criterion used to define theoretical saturation was the redundancy of major themes identified in subsequent interviews, that did not provide any major additional information.

Transcripts were not returned to participants, but two feedback focus groups were organized with those who accepted, during the first interview, to be informed about the results of the research, providing them the opportunity to suggest modifications and comments. During those meetings, participants confirmed the results of the research, as well as the prioritization and organization of themes and sub-themes. Additional insights about the studied phenomenon were evoked by participants during these focus groups, which were analyzed and then integrated in the results by GB.

## Results

Between January and March 2021, a total of 26 people were included in the study: 10 pregnant women, 10 women in the postpartum period, and six people from their social surroundings. Their socio-demographic characteristics are detailed in Table 3. At the time of recruitment, eight people were living in homeless hotels, seven in slums, five in healthcare institutions (maternity department), two in homeless shelters, two in apartments provided by an association, and two in apartments owned by friends or family members. More than one participant in the study has declared to have lived rough at least once during pregnancy. During recruitment, four people refused to participate in the study as they were not interested or did not have time for the interview. Even though the opportunity to interrupt the interview at any moment was explained by the interviewer during recruitment, no one among the participants included in the study have requested this.

Three main themes were outlined from the analysis of interviews: i) the overall perceived negative impact of homelessness on all aspects of health (physical, mental, and social), ii) the positive influence of social and professional support on health during pregnancy and the postpartum period, and iii) having access to suitable housing, being independent, and being in good health as the main priorities of the study sample. All themes and sub-themes and their organization can be found in the coding tree in S1 Fig.

### Theme 1: The overall perceived negative impact of homelessness on all aspects of health (mental health, physical health, and social well-being)

Even though the living conditions included in the definition of homelessness set out in this study are very heterogeneous, a common perception of an overall negative impact of homelessness on all aspects of health emerged from the interviews.

**Mental health: Stress, anxiety, and phycological suffering.**   Firstly, mental health was perceived as the aspect most impacted by this phenomenon. Women during pregnancy or during the postpartum period reported constantly feeling stressed, anxious, and psychologically unwell, up to the point of not being able to sleep, to rest, or to think about something else beside their actual situation. More specifically, the factors identified by participants as responsible for this perception were:

- Uncertainty about the future.
  Uncertainty about the future derived mainly from housing instability. The risk of having to leave their living place at any moment, impeded women to project themselves into the near future. The absence of a perspective including a stable living condition stressed and worried them, as they feared to find themselves and their children roofless.

  *"You see, we live in a hotel. . . The hotel is not an apartment where we are going to stay, it is a hotel. . . We don't know, at any moment it can change. At any moment they can say that the contract is finished. But then, where are you going to go with this child? It's like you're there, but you're not relaxed, because you don't know when you can be told to leave. So, it is a bit stressful"* [Participant 6]

- Irregular administrative status.
  Not having a regular administrative status was another factor related to stress and to fear. Participants felt unsafe during their daily life because of the fear of having their documents checked and of having troubles with the local authorities (being detained or expelled from the country).

*"The fact of being in an irregular situation, if you are controlled, well. . . you will be in trouble with the authorities. So mentally it's complicated because you're in the street, you walk but you don't feel safe. . . because if there's a control you don't have the documents. . . Here, to be really at ease, you must have the documents, you must have a regular administrative status. [. . .] Stability means having a regular administrative status"* [Participant 8]

- Financial precarity and dependence on others.
  The financial precarity experienced by the study population was often pointed out as the main cause of their dependence on others (mainly family members or friends). The perpetuation of this condition impeded women to feel at ease in their living place and negatively impacted their psychological status.

*"Health for me means having a roof over my head, an income. Here, in this place, it's the lady (sister-in-law) who does almost everything for me, and this makes me feel uncomfortable. So, I'm (psychologically) not well. I'm dependent on the lady for everything."* [Participant 25]

- Inadequate living conditions associated with homelessness.
  Women often experienced extreme living conditions, such as discomfort, low hygiene, and difficult access to water, electricity, and heating. These conditions were highlighted as responsible for the many problems encountered during everyday life and for the constant stress perceived by the study population. Furthermore, pregnant women were particularly concerned when thinking about taking care of a newborn in such conditions.

*"It's not easy at all when you live in an uncomfortable and inadequate place. So, you're stressed all the time, you have problems all the time, and it's not straightforward at all."* [Participant 5]

*"I'm worried, because when you have a little baby, you must have water, hot water, electricity, and clean places."* [Participant 15]

- Barriers to healthcare access.
  Barriers in obtaining a health insurance, due to lack of information and to complex administrative procedures, and the consequent difficulties to access the healthcare system, were evoked as source of anxiety.

*"I don't have health insurance to treat me. I don't know how it works and I don't know how to apply for it. So, I feel anxiety because of this and because of my health status."* [Participant 19]

*Social well-being*: *Social isolation*. Social isolation was identified during interviews as one consequence of homelessness and, simultaneously, as a factor which had a negative impact on mental health. Overnight shelters, homeless hotels, temporary accommodation, and slums, where participants were living at the time of the study, did not have enough space to allow people to invite friends and family members, and were often far away from city centers, thus creating a geographical barrier to social integration. Not being able to meet people and to learn social norms and the local culture has been described by participants as having a negative impact on social integration and, consequently, on mental health.

*"So, here you are alone in your apartment, in your thing, and if you don't meet other cultures, it's not good for your integration, nor for your mind, nor for your children. You are not going to learn the culture and, even psychologically, you are not going to feel good in this country."* [Participant 20]

**Physical health: Food insecurity, physical suffering, aggravation of chronic conditions, pregnancy complications, and risk of infections.** During interviews, food insecurity was the second most quoted consequence of homelessness on health. Women perceived their diet as significantly poor, as they were skipping meals on a regular basis and spending an important part of the day looking for food. Various factors were recognized as being behind this feeling:

- Lack of cooking and food storing facilities in most homeless shelters and hotels.
Most of homeless shelters and hotels were not equipped with any cooking and food storing facilities, forcing women to appeal to friends and family members to find a place where to cook. Having to constantly go at somebody else's houses to cook was described as even more complicated during pregnancy and the postpartum period, mainly due to physical conditions. In addition, family and friends were not always available to provide a kitchen to cook.

*"The 115 (the emergency accommodation number) has put us in a place where I cannot cook, so I must go and cook at somebody else's house and, given my condition, it is complicated to go out all the time to cook. If I want to eat, I must go out. [. . .] It's complicated to go to somebody else's house all the time. There are days when the person doesn't want to see you, but, even so, the need is there, and you must go (to cook there). There are also days when, physically, I don't feel well, so I stay at my place. I don't go out and I don't eat."* [Participant 3].

The absence of a kitchen and the fact of skipping meals was described by participants to the study as having an impact on mental health.

*"Having a place, where I can cook and that's it, psychologically it will help me. But the fact that I have this constraint. . . It's complicated to stay without eating sometimes"* [Participant 3]

- Financial precarity.
Financial precarity, along with the absence of a kitchen in their living place, pushed women to appeal to food distribution organized by associations directly in homeless hostels and shelters. However, women were confronted to a limited food choice, often of low quality, incapable of meeting dietary and cultural needs.

*"The food they offer me there (food distributions) it's not healthy. I have a hard time eating that, and, since I don't have an income, I can't buy food to eat properly."* [Participant 19]

*"I had gestational diabetes, and the food they gave me at the hotel was not adapted to my illness."* [Participant 5]

*"Sometimes I get a headache from going to bed without eating anything and I don't have money on me to go buy something else to eat, so I have to eat what they give me there (at the homeless hotel)."* [Participant 17]

*"Sometimes there's this food they bring to us that we're not used to eat. . . We can't eat it."* [Participant 4]

Lastly, physical suffering and pain, aggravation of chronic diseases, higher risk of infections, and pregnancy complications were outlined by participants as consequences of different factors related to homelessness:

- Inadequate living conditions associated with homelessness.
Difficult access to water, electricity, and heating in slums and squatter settlements, resulting in low hygiene and exposure to cold temperatures, were not only identified as source of constant stress, but of physical suffering, higher risk of infections, and aggravation of chronic conditions.

*"The most difficult thing is to do the laundry, because there is no water, and you must get the water for yourself. It is especially when you need to lift the weight (of the water to carry it) that you feel pain in the stomach."* [Participant 2].

*"If you put a piece of bread on the table, they (the insects) walk everywhere and you are not going to eat it, they go on the bread and then you can get diseases and other things."* [Participant 9]

*"The environment can affect my health or the health of my child. Because my child has an asthma problem. If we try to clean, it is always going to be dirty because we are in a slum, and it is also cold."* [Participant 9]

- Pregnancy complications due to psychological stress and fatigue.
Finally, the stress and anxiety perceived by women because of the constant uncertainty about the near future due to housing instability, were recognized as responsible for the pregnancy complications experienced (preterm birth).

*"All this stress is the cause of everything I am going through today. The fact of asking myself all the time "where am I going to sleep tonight?", "what is going to happen tonight?". All this in the head, while being pregnant, is not easy. This is what made me almost deliver at four months and a half of pregnancy."* [Participant 5].

### Theme 2: The positive influence of social and professional support on health during pregnancy and the postpartum period

**Social support.** Having a strong social environment was seen as fundamental to be in good health during pregnancy and the postpartum period. The relationship of women with their social surroundings, especially with family and friends, was described as a helping relationship on different levels:

- Help during daily activities and tasks, such as childcare and housework.

*"It feels good when you have someone. Because, sometimes, when you have children and you don't have anyone to help you to take care of them, it's not easy to live. Sometimes, when my child is not at daycare and I have an appointment, I can call my husband so that he keeps the child, and I can go to the appointment.* [Participant 15]

*"It is my husband who helps to go to buy groceries, to cook, to do the laundry."* [Participant 7]

- Moral support.

*"It helped me because she (her friend) supported me, she told me: "It will pass, it is difficult now, but you will see it will get better". She gave me courage so that I could rest."* [Participant 19]

- Accompaniment to medical and administrative appointments.

*"Sometimes I went alone (to medical and administrative appointments), sometimes I went with a friend who spoke French and who helped me. She used to translate when it was complicated for me to understand."* [Participant 25]

- Housing.

*"I met her at the train station because I was sleeping rough for 2 days. I was pregnant and when I saw her and heard her speaking, I knew that she was from the same country as me, so I approached her and told her my problem and she hosted me in her house, gave me food, and a place where I could shower and sleep."* [Participant 5]

**Professional support.** The "welcomeness" of healthcare professionals during pregnancy follow-ups was positively valued by most of the women included in the study, as well as their social surroundings. Indeed, being informed by healthcare professionals and having the opportunity to ask questions about topics related to pregnancy, childbirth, and the newborn's wellbeing was described as helpful.

*"In general, it went very well at the end of the medical appointments. If you have any questions, the nurses, the doctors, or the assistant explained it to you very well."* [Participant 21]

*"Yes, they are attentive. There is even a social worker who takes good care of me, and the caregivers are welcoming as well."* [Participant 10]

*"The hospital it was really helpful for me. They really tried and they help me a lot. So, it was very stress-free. I would come to appointments, I would see a doctor, the doctor would check me, check my baby, you know. . ."* [Participant 22]

Moreover, participation in discussion groups, organized by healthcare professionals working in the maternity department of the CHU of Nantes, was perceived positively by those who had the opportunity to attend them. More specifically, these activities allowed women to share information, expertise, and doubts with the group.

*"They taught me a lot of things because they subscribed me in a group of baby and moms (mothers having given birth for the first time in France). They explained everything that happens in pregnancy. How to have a baby, how to behave, what to eat or not, etc."* [Participant 18]

### Theme 3: Having access to suitable housing, being independent, and being in good health as the main priorities of the study sample

The priorities of people included in this study were (in order of importance): i) having access to suitable housing, ii) being independent, and iii) being in good health. Often these three elements were mentioned at the same time and linked to each other.

Firstly, participants described suitable housing conditions as sufficiently spacious, furnished with a kitchen, and equipped with a proper access to essential services, such as water, electricity, and toilets.

*"I would like to have a mobile-home to have an accommodation [. . .] Because in mobile-homes, there is water, there are showers, there are toilets, you have a kitchen."* [Participant 12]

*"For me (the priority) it's having enough space. I can isolate myself, but I can't stay so much. . . so much like that. But when I have space, I can receive people [. . .]"* [Participant 18]

Secondly, the concept of being independent was described as a prerequisite to finding a role in society and to giving a proper future to family and especially kids. Independence was directly connected with having access to a job and to a regular administrative status, the latter being identified as the fundamental initial step in the whole process.

*"The administrative status is the everything in fact. When you don't have a regular administrative status, you can't work, you can't do anything. It keeps me awake sometimes, so much that I think about it in my head"* [Participant 26]

*"I think it is mainly the opportunity to work so that I can give a good future to my daughter. Because I think that everything depends on me, and I don't want my daughter to be born and to continue living in a hotel. So, if I could change something it would be the opportunity to work or to have a training so that I could work and give a good future to my daughter"* [Participant 15]

Lastly, being in good health was linked with having access to healthcare coverage and, therefore, to the healthcare system when needed.

*"Apart from the administrative status, and housing, there is health: going to the doctor without restriction. These are the three fundamental things, and the rest will come later. If we have these three things, we are a bit settled."* [Participant 26]

## Discussion

This study aimed to provide a general overview of the perceived impact on health of the physical, social, and healthcare surroundings of homeless women during pregnancy and the postpartum period, prioritizing women's perspective. One-to-one semi-structured interviews allowed us to identify three main themes regarding the study objective: i) the overall perceived negative impact of homelessness on all aspects of health (physical, mental, and social), ii) the positive influence of social and professional support on health during pregnancy and the postpartum period, and iii) having access to suitable housing, being independent, and being in good health as the main priorities of the study sample. A comparison of these results with those issued from previous quantitative and qualitative research, focusing on specific health-related problems and dimensions of homelessness, has helped us to further explore the different themes and sub-themes evoked by the study sample.

Interviewees perceived an overall negative impact of homelessness on all the dimensions of health (physical, mental, and social) during pregnancy and the postpartum period. These consequences were often connected one to another and had common determinants, underlying, once again, the complexity of homelessness and of its interactions with health. A particular

accent on mental health was evoked by women, who frequently reported feelings of stress, anxiety, and concern. These feeling were documented also by a meta-synthesis of qualitative studies on homeless mothers, where suicidal thoughts and depression were also identified as consequences of homelessness [14]. High prevalence of major depressive disorder among mothers experiencing homelessness has been documented and significantly associated with life-time suicide risk, post-traumatic stress disorder, and unmet health needs [15, 16]. Food insecurity during pregnancy and the postpartum period was another major sub-theme in our results, and was connected by participants to psychological distress. Limited food choice, poor nutrient intake, disrupted eating patterns, and skipped meals (also affecting children) in homeless families and pregnant women has been related to lack of income, inadequate or lack of cooking and food storage facilities, unhealthy food supply distribution, housing instability, social isolation, and depression [17–19]. These results are relevant as food insecurity during pregnancy has been associated adverse perinatal outcomes, breastfeeding interruption, and maternal stress, anxiety, and depression [20]. Lastly, inadequate living conditions associated with homelessness were related to physical suffering, higher risk of infections, and aggravation of chronic conditions. Low levels of water sanitation and hygiene during pregnancy have been associated with higher risks for health, such as infections, as well as with increased maternal mortality [19, 21]. In our study, adverse perinatal outcomes, and more specifically preterm birth, were identified as a consequence of the stress perceived because of homelessness. Multiple epidemiological studies have explored perinatal outcomes in homeless women, underlying a significant association between homelessness and preterm birth, low birthweight, neonatal intensive care unit admissions, and delivery complications [11].

During interviews, social and professional support were positively valued by interviewees. Different forms of help, such as help during daily activities and tasks, moral support, accompaniment to medical and administrative appointments, and housing were related to social support during interviews. Being supported by family members, friends, and other people experiencing homelessness has been recognized as having a positive influence in accessing the healthcare system and in reducing stress levels, depressive symptoms, and feelings of isolation and marginalization in homeless mothers [14, 22–24]. The quality of professional support was related to the "welcomeness" and the availability of health professionals during pregnancy follow-ups, as well as the opportunity to interact with other mothers facing the same situation and challenges. However, most of the published scientific material exploring the relation of homeless women with healthcare providers, has described interactions with healthcare professionals as a barrier in accessing antenatal and postnatal care in different situations [22]. These contrasting results could derive from the fact that most of the women interviewed in our study were followed during their pregnancy by a specific unit of the maternal department of the CHU of Nantes, specialized on following pregnant women experiencing precarity. On the other hand, organized peer support in pregnancy and early parenthood has been described as a promising tool in reducing anxiety, feelings of isolation, disempowerment, and stress, while improving self-esteem, self-efficacy, and parenting competence [22, 25].

The qualitative approach adopted in this study permitted women to spontaneously evoke the themes that mostly mattered to them within each pre-identified category. The main priorities of the study sample (suitable housing, independence, and good health) have been previously classified as basic needs among the homeless population, and were listed among the prerequisites for health listed in the Ottawa Charter for health promotion, signed by WHO in 1986, as well as in the 25[th] article of the Universal Declaration of Human Rights [26–28]. We used these three axes to explore possible solutions to the consequences of homelessness on health during pregnancy and the postpartum period.

Adapted housing was identified as the principal priority of the study population and could represent the first fundamental step in tackling this issue. Housing interventions, such as the Housing First model and supportive housing, identify access to suitable and stable housing for homeless people as its core principle [29]. Supportive housing has been described as effective in reducing homelessness and achieving housing stability for people with moderate and high support needs when compared to usual care [30]. The Housing First program had shown promising health, employment, housing, and social outcomes in homeless pregnant/parenting women using substances [31]. However, due to low standardization in outcome measure domains and tools, further research is needed to correctly assess the long-term effects of housing interventions on physical and mental health, as well as on quality of life [30, 31]. On the other hand, income assistance interventions, such as housing subsidies, have been described as capable of improving housing stability, independent housing, quality of life and food security [30].

The second priority of the interviewees was to gain independence and to integrate into society. Employment has been described to play a major role in the social integration process of homeless people, as it is an essential step in gaining financial independence and in creating a sense of belonging to the community [32]. Different forms of interventions promoting employment, training, and education of homeless people have been suggested as playing an important role in exiting homelessness [33]. These interventions could represent valid solutions also for homeless mothers and homeless pregnant women. Moreover, social support provisions for homeless mothers have been related better expectations for employment, as well as better physical and mental health outcomes [32]. The development of a social network, along with all its benefits, can be stimulated through group activities, such as workshops, discussion groups, and, more widely, recreational and educational activities. Participatory women's groups are listed by WHO among their recommendations on antenatal care for a positive pregnancy experience, as they represent an opportunity to increase mutual support between pregnant women [34].

Finally, having access to healthcare and being in good health were also a priority of the study population. Poor engagement with antenatal care was reported among homeless women, and it was related to different barriers in accessing healthcare, such as lack of person-centered care and complexity of survival while being homeless [22]. Moreover, the negative consequences of homelessness on health are complex and interconnected, requiring, therefore, to be addressed by multidisciplinary structured approaches. Outreach services capable of promoting collaborative networks with other service providers have been described as promising models to address the complex health and social needs of pregnant homeless women [35]. Primary health programs integrating physical, mental, and social services have been associated with positive outcomes in the homeless population, such as change in social status and housing status, and improvement in access and satisfaction in the use of healthcare services [36].

The results of this study should be interpreted considering the following limitations. The COVID-19 pandemic strongly influenced data collection. Confidence and trust with participants were sometimes hard to establish due to the strict observance of barrier and physical distancing measures during interviews, as well as due to the feeling of distrust towards institutions sometimes expressed by the study population and documented also in previous research [37]. Moreover, the fact that interviews were organized during a single meeting could have had an influence on the exploration of certain sensible themes related to homelessness and documented in previous research, such as sexual violence and addictions. Finally, the inclusion of non-French speakers and the use of a professional interpreter, allowed a reduction in risks of selection bias, but increased probabilities of interpretation and translation bias. However, the research team has tried to limit these probabilities by organizing briefings and debriefings with professional interpreters, and by following published recommendations [38].

## Conclusions

This qualitative study provides a comprehensive view of the perceived impact of homelessness on the health of women, living in the metropolitan area of Nantes, during pregnancy and the postpartum period, and on the factors associated with this perception.

Firstly, an overall negative impact of homelessness on mental health, physical health, and social well-being was described by participants in the study. Social support and the "welcomeness" of healthcare professionals during pregnancy follow-ups were identified as positive factors, helping women to cope with their actual situation. Basic needs, such as suitable housing, being independent, and being in good health were the main priorities listed during interviews by the study sample. Coherent findings were found when comparing these results with those issued from past studies focusing on specific health outcomes or dimensions of homelessness.

Secondly, using women's perspectives and priorities, different axes were identified as possible solutions in tackling homelessness and its consequences on health during pregnancy and the postpartum period. Housing interventions, such as the Housing First model and supportive housing, and interventions promoting employment, training and education have shown to be promising ways forward in helping people to exit homelessness and in improving their health, social integration, and independence. However, evidence concerning these interventions in homeless pregnant women and women in the postpartum period is still limited and further research is needed to assess short- and long-term outcomes in this specific population. In the meanwhile, a multidisciplinary structured approach in healthcare is needed to limit the vast consequences of homelessness on health during pregnancy and the postpartum period, improving access to services capable of meeting the complex health and social needs of this population.

## Supporting information

**S1 Table. COREQ (COnsolidated criteria for REporting Qualitative research) checklist.**
(PDF)

**S1 Fig. Coding tree.**
(PDF)

## Acknowledgments

The authors are thankful to the study participants to have agreed to share information and their perspective. The authors acknowledge Claire Dubois, Marie Laluque, Houda Merimi, and Katell Olivier for their support in study conceptualization and supervision. The authors thank the staff of the Berekty and Asamla associations for the translations provided during data collection. The authors thank the "Les Forges Médiation" association and the maternity department of the CHU of Nantes for helping with the recruitment of participants.

## Author Contributions

**Conceptualization:** Giulio Borghi, Pascal Caillet, Sylvaine Devriendt, Maxime Lebeaupin, Maud Poirier, Juan-Diego Poveda.

**Data curation:** Giulio Borghi.

**Formal analysis:** Giulio Borghi.

**Investigation:** Giulio Borghi.

**Methodology:** Giulio Borghi, Pascal Caillet, Maxime Lebeaupin, Juan-Diego Poveda.

**Project administration:** Giulio Borghi, Sylvaine Devriendt.

**Supervision:** Sylvaine Devriendt, Juan-Diego Poveda.

**Validation:** Pascal Caillet.

**Visualization:** Giulio Borghi.

**Writing – original draft:** Giulio Borghi.

**Writing – review & editing:** Pascal Caillet, Sylvaine Devriendt, Maxime Lebeaupin, Maud Poirier, Juan-Diego Poveda.

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
