## [Decision Letter · Decision Letter 0]

26 Sep 2022

PONE-D-22-17988The perceived impact of homelessness on health during pregnancy and the postpartum period: a qualitative study carried out in the metropolitan area of Nantes, FrancePLOS ONE

Dear Dr. Dr. Giulio Borghi,,

Thank you for submitting your manuscript to PLOS ONE. After careful consideration, we feel that it has merit but does not fully meet PLOS ONE’s publication criteria as it currently stands. Therefore, we invite you to submit a revised version of the manuscript that addresses the points raised during the review process.

We look forward to receiving your revised manuscript.

Kind regards,

Zemenu Yohannes Kassa, Msc

Academic Editor

PLOS ONE

Journal Requirements:

2. In the ethics statement in the Methods, you have specified that verbal consent was obtained. Please provide additional details regarding how this consent was documented and witnessed, and state whether this was approved by the IRB.

Additional Editor Comments:

Dear Dr. Giulio Borghi,

Academic editors’ comments

The topic of the manuscript is interesting. Nevertheless, the reviewers raised several concerns: considering this point, I invite authors to perform the required major revisions.

#Abstract

Introduction

1. Line 24-25 your objective seems quantitative study, you should change it.

# Introduction

1. Line 47-49, you should insert the reference.

2. You should strictly follow the guideline Plos one on how to put citations in the manuscript.

For example in your manuscript line 50……. has been described in the multiple EU Member States.[1].Change it like this ….. been described in the multiple EU Member States[1].

3. line 51, you showed your reader in 2012 data, it is too late, please show us the latest one number no more than five years data.

#. Method

1. In line 75, did you use any materials? If not, please remove the materials.

2. In your qualitative study, which approaches you used and explain in detail? The way of your data collection methods like IDIs or FGD?

3. I am wondering if you explain briefly how to select the study participants, whether they have mental problems or not, and are under the age of 16 years old.

4. In line 116, in the qualitative study leading questions does not recommend while you’re your questionnaire is the leading question. Are there any incentives for study participants for time compensation?

5. line 215, in the principle of ethics healthcare providers, do not act as data collectors on patients.

#Discussion

1. line 244-247 it doesn’t similar to your study population. you should remove it. Your study population is pregnant and postpartum mothers who are homeless, but not all women.

Therefore, you should focus on this population and your discussion needs gross modifications.

Reviewers' comments:

Reviewer's Responses to Questions

**Comments to the Author**

1. Is the manuscript technically sound, and do the data support the conclusions?

Reviewer #1: Yes

Reviewer #2: No

2. Has the statistical analysis been performed appropriately and rigorously? 

Reviewer #1: Yes

Reviewer #2: I Don't Know

3. Have the authors made all data underlying the findings in their manuscript fully available?

Reviewer #1: Yes

Reviewer #2: No

4. Is the manuscript presented in an intelligible fashion and written in standard English?

Reviewer #1: Yes

Reviewer #2: Yes

5. Review Comments to the Author

Reviewer #1: The manuscript describes a technically sound piece of scientific research with data that supports the conclusions.

The statistical analysis been performed appropriately and rigorously.

All data underlying the findings in the manuscript is fully available.

The manuscript presented in an intelligible fashion and written in standard English. The language in submitted articles must be clear, correct, and unambiguous.

(Spelling considerations)

INTRODUCTION - “a comprehensive view of this phenomenon, prioritising” (PRIORITIZING)

MATERIALS AND METHODS - “ and to prioritise the perspective” (PRIORITIZE)

MATERIALS AND METHODS - “Recruitment was organised” (ORGANIZED)

… (the same applies for organisation X ORGANIZATION, analysing x ANALYZING, centre X CENTER, standardisation x standardiZation,

Theme 1: The overall - “facilitiesin most” (FACILITIES IN)

Reviewer #2: While I can appreciate the value of gaining perspectives of this study’s participants, major revisions are recommended.

INTRODUCTION

Lines 49-51: State the number of homeless persons in France in 2012 to make this statement clearer and have a greater impact on the severity.

Lines 69-70: It is not accurate to say that this data is missing. You mention many studies in your discussion section. Furthermore in my brief search I found more studies worth referencing / reviewing on this topic that were not included in your references:

- https://doi.org/10.1016/j.ijnurstu.2021.103974

- https://doi.org/10.1111/hsc.12972

MATERIALS/ METHODS

Lines 77-78: Describe the COREQ in more detail to make your methods and results more clear. This section should have enough detail to allow others to replicate your study.

PARTICIPANTS AND RECRUITMENT

Line 84: Share some examples of who people “from their social surroundings” are in relation to the participants. Including other participants strays away from your stated study objective to highlight the voices and opinions of the women themselves. It takes away from these women to include voices of others within this study.

Lines 87-91: I would make a statement on how (/if) the use of ETHOS effected your recruitment.

Line 96: What is the inclusion criteria?

DATA COLLECTION

Lines 128-129: Did your professional interpreters assist in the transcribing process to ensure that the true messages of participants were not changed in translation? If so, I would state that here and in the data analysis section.

DATA ANALYSIS

Overall more detail is needed here. What were the pre-identified categories? What themes and subthemes emerged during data collection?

Line 44: state any suggestions, modifications, or comments made by participants in the focus groups. How did you use these suggestions to shape your results/ themes?

RESULTS

Lines 148-152: I would organize the results into the ETHOS definition mentioned earlier.

Lines 163-164: consider integrating quotes of participants into the themes below since your objective is to highlight their voices and perspectives. This would make the themes/ results more powerful and impactful to read

THEME 1: Recommend creating subsections of each area of health (physical, mental, social)

THEME 2 + 3: There is not much data in either of these sections. Perhaps consider combining these to something similar to a “personal and professional social supports” theme, or add quotes as mentioned earlier to justify why these two themes are widely different.

THEME 4: again, it seems like there is not much data in this section. It may make sense to combine this into theme 1 and make a comment that these are the priorities to the participants. This can be further emphasized in the discussion/ conclusions sections,

DISCUSSION:

I understand the reason to bring up previous research here to back your own, but you did not describe how your article adds any thing different to the current body of research on this topic. In one section you claim this research is missing in current literature, but then you give a very lengthy discussion on all of the research that is similar. Consider restructuring this to focus on what your research uniquely adds that these articles do not.

Lines 261-262: what were the adverse perinatal outcomes your participants experienced?

Line 266: what are the different forms of help.

CONCLUSIONS:

this section needs more details, examples and research to justify your conclusions. What are the next steps and how do they relate to your results given your participants’ priorities. An idea would be to structure your discussion based on their top priorities and the literature says may help in supporting their priorities.

6. PLOS authors have the option to publish the peer review history of their article (what does this mean?). If published, this will include your full peer review and any attached files.

Reviewer #1: **Yes: **Estevao Cubas Rolim

Reviewer #2: **Yes: **Alyssa Fabianek, OTD, OTR/L, CBIS

---

## [Author Response · Author response to Decision Letter 0]

8 Nov 2022

To editor:

Dear editor, the authors are deeply thankful for your consideration and for the interest you have shown for our work. We have addressed below the points raised during the review process and we have corrected the manuscript as suggested. 

Best regards,

Dr Giulio Borghi

1. We have ensured that our manuscript meets PLOS ONE’s publication criteria.

2. We have provided additional details about how oral consent was obtained in lines 108-111. 

3. The minimal data needed to support our study results have been integrated in the “Results” section of the manuscript. Nevertheless, as detailed in the data availability statement, the full interview transcripts are not available for the following ethical reasons: 

• The interview transcripts contain sensible information that, when crossed, could easily lead to the identification of participants (dates, names, addresses, specific life events, country of birth, languages spoken). 

• The relatively small group of homeless women being pregnant or in the postpartum period in the metropolitan area of Nantes, could easily lead to the identification of participants.

• The de-identification or anonymization of the transcripts is not possible due to the qualitative approach adopted in this research. As a matter of fact, the unique life stories shared by participants could easily lead to their identification if considering the geographical area included in the study. 

We assure you that none of the unacceptable data access restrictions cited in PLOS ONE’s guidelines apply to our case. 

Additional editor comments:

# Abstract

1. The study objective was changed as follows: “The objective of this study was to describe the overall perceived impact of homelessness on health during pregnancy and the postpartum period, prioritizing women’s perspective through a qualitative research approach.” (lines 24-16).

# Introduction

1. We changed the citations’ format following the Journal’s guidelines. 

2. More recent data has been inserted in the manuscript as requested. The new sentence is the following “In the specific case of France, the last estimations made by the French Court of Auditors, had shown an increase in homelessness of nearly 110% between 2012 (143,000 people) and 2019 (300,000 people).” (lines 48-50).

# Method

1. Materials was removed as requested. 

2. All the information of the qualitative approach used in our study can be found in the pages listed in the “study design” and “analysis and findings” sections of the COREQ checklist that we have added in the supporting information (S1 Table). Data was collected through one-time individual semi-structured interviews, as detailed in line 113. 

3. The explanation on how study participants were selected in respect to age and mental health problems has been detailed in lines 83-88. 

4. No incentives for study participants for time compensation were offered. 

5. The positive evaluation of the discussion groups organized by the CHU of Nantes was freely evoked by participants during the interview. Our study did not aim to evaluate the satisfaction of those discussion groups. Nevertheless, interviewees were able to share their pregnancy follow-up experience without any restriction, either if it was positive or negative, and to share the experiences that mattered the most to them. 

#Discussion

1. Gross modification and restructuring of the discussion have been done, focusing on the study population, as suggested. 

To reviewers: 

Dear reviewers, all the authors deeply appreciated the interest you have in the topic. We are thankful for the interesting insights you have highlighted in your reviews, as they allowed us to improve the quality and clarity of our manuscript. We have addressed below each of the points you have evoked during your review. 

Best regards,

Dr Giulio Borghi

Reviewer #1:

All the grammatical corrections suggested have been made. 

Reviewer #2:

INTRODUCTION

1. The number of homeless persons has been detailed as requested. The last available estimations were inserted as recommended by the editor in comment #1 in the introduction section. The new sentence is the following: “In the specific case of France, the last estimations made by the French Court of Auditors, had shown an increase in homelessness of nearly 110% between 2012 (143,000 people) and 2019 (300,000 people).” (lines 48-50).

2. The previously published research focuses on specific health outcomes (e.g., perinatal outcomes, food insecurity, mental health, etc.), on specific dimensions of homelessness (e.g., housing instability), or on a different population (e.g., overall homeless population). On the other hand, our research aims to provide a general description of the overall perceived impact of homelessness on health during pregnancy and the postpartum period, without focusing on a specific dimension of homelessness or on a specific health outcome, while prioritizing women’s perspective. We agree that data on the topic is not missing. We rather meant that a general description of this phenomenon from women’s perspective is not available yet and that it would be a helpful perspective in identifying possible solutions. For these reasons, we corrected the sentence (lines 65-67).

The study (https://doi.org/10.1111/hsc.12972) focusing on perceived barriers and facilitators in antenatal and postnatal healthcare access by homeless women has been integrated as suggested. On the other hand, the other reference (https://doi.org/10.1016/j.ijnurstu.2021.103974) was not cited as it focused on healthcare access of homeless women in general, and as we already included studies on this topic concerning our study population. 

METHODS

1. The COREQ checklist was added in the supporting information (S1 Table). The checklist indicates in which page of the article each specific methodological aspect is provided. 

PARTICIPANTS AND RECRUITMENT

1. Examples of who are people “from their social surroundings” have been added in lines 83-88. The choice of integrating people from women’s social environment was made to promote a global vision of the perceived impact of homelessness on the health of the study sample, and to reinforce the robustness of the results by cross-referencing the data collected during interviews. Nevertheless, this data was used as complementary material to further explore the studied phenomenon, being careful to respect women’s priorities and perspective. Moreover, as our research aimed at exploring how the social environment was perceived to impact the health of homeless women during pregnancy and the postpartum period, having the opportunity to interview people from their social surroundings helped us to better understand this phenomenon. 

2. Details on how the use of ETHOS affected the recruitment of participants have been added in line 101-103.

3. Inclusion criteria have been detailed in lines 83-88. 

DATA COLLECTION

1. The professional interpreters did not participate in the transcribing process. However, as the translation took place at directly during the interview, the interviewer made sure that the messages translated by the interpreters were correct by reformulating the answers in order to ask confirmation to participants directly during interviews. Moreover, results were confirmed by participants during the focus groups organised with those who accepted, during the first interview, to be informed about the results of the research. 

DATA ANALYSIS 

1. The coding tree has been added in the supporting information to provide further insights about data analysis (S1 Fig). 

2. Further details about how the focus groups were integrated in the results have been added in line 159-161. 

RESULTS

1. The dimensions of homelessness included in the ETHOS classification, as well as other factors related to them, were included as the final sub-themes, being the determinants of the health status perceived by women. We have prioritized a classification of results based on our study objective, which focused on women’s perception on how their health was impacted by homelessness. Moreover, some dimensions of homelessness, such as housing instability, could have an impact on more than one health dimension, creating difficulties on classifying the different perceptions of women. The inclusion of the coding tree in the supporting information provides a better view of the organization adopted and on the rationale behind this choice. 

2. Quotes from participants have been added to the text as suggested. 

3. Subsections of each dimension of health (mental, physical, social) have been created and the presentation of results has been rearranged accordingly. 

4. Theme 2 and 3 have been combined as suggested. 

5. Despite theme 4 contains less data than the others, we have chosen to keep it divided to emphasize the priorities of the study sample. The integration of the priorities in other sub-themes could reduce their relevance. Moreover, the topics highlighted as priorities could have a benefit on more than one dimension of health, making their classification under one theme difficult. We preferred to keep this theme because we used it in the discussion to explore possible solutions.

DISCUSSION

1. The comparison of our results with previously published research was done to further explore the themes evoked during interviews by adding insights found by previous research on that specific theme. This allowed us also to have a general picture on how the perception of the concerned population and of their priorities is aligned with previous research. The second part of the discussion was based on women’s priorities, exploring possible ways forwards, as suggested. 

2. The adverse perinatal outcomes evoked as a consequence of homelessness have been detailed in lines 412-415. 

3. The different forms of help have been detailed in lines 417-420. 

CONCLUSIONS 

1. The discussion has been restructured. In the conclusion section we provided a general resume of the findings and implications of this research.

---

## [Decision Letter · Decision Letter 1]

26 Dec 2022

The perceived impact of homelessness on health during pregnancy and the postpartum period: a qualitative study carried out in the metropolitan area of Nantes, France

PONE-D-22-17988R1

Dear Dr. Giulio Borghi,

We’re pleased to inform you that your manuscript has been judged scientifically suitable for publication and will be formally accepted for publication once it meets all outstanding technical requirements.

Kind regards,

Zemenu Yohannes Kassa, Msc

Academic Editor

PLOS ONE

Additional Editor Comments (optional)

Reviewers' comments:

Reviewer's Responses to Questions

**Comments to the Author**

1. If the authors have adequately addressed your comments raised in a previous round of review and you feel that this manuscript is now acceptable for publication, you may indicate that here to bypass the “Comments to the Author” section, enter your conflict of interest statement in the “Confidential to Editor” section, and submit your "Accept" recommendation.

Reviewer #1: All comments have been addressed

2. Is the manuscript technically sound, and do the data support the conclusions?

Reviewer #1: Yes

3. Has the statistical analysis been performed appropriately and rigorously? 

Reviewer #1: Yes

4. Have the authors made all data underlying the findings in their manuscript fully available?

Reviewer #1: Yes

5. Is the manuscript presented in an intelligible fashion and written in standard English?

Reviewer #1: Yes

6. Review Comments to the Author

Reviewer #1: The manuscript describes a technically sound piece of scientific research with data that supports the conclusions.

The statistical analysis been performed appropriately and rigorously.

All data underlying the findings in the manuscript is fully available.

The manuscript is presented in an intelligible fashion and written in standard English.

7. PLOS authors have the option to publish the peer review history of their article (what does this mean?). If published, this will include your full peer review and any attached files.

Reviewer #1: No

---

## [Editor Report · Acceptance letter]

2 Jan 2023

PONE-D-22-17988R1 

The perceived impact of homelessness on health during pregnancy and the postpartum period: a qualitative study carried out in the metropolitan area of Nantes, France 

Dear Dr. Borghi:

I'm pleased to inform you that your manuscript has been deemed suitable for publication in PLOS ONE. Congratulations! Your manuscript is now with our production department. 

Kind regards, 

on behalf of

Dr. Zemenu Yohannes Kassa 

Academic Editor

PLOS ONE